

# Genome sequence analysis of Malayan pangolin (*Manis javanica*) forensic samples reveals the presence of *Paraburkholderia fungorum* sequences

Ka Yun Tan[1,*], Siwei Deng[2,*], Tze King Tan[3], Ranjeev Hari[1], Frankie Thomas Sitam[4], Rofina Yasmin Othman[5], Kum Thong Wong[6], Taznim Begam Mohd Mohidin[1] and Siew Woh Choo[2,7,8]

[1] Institute of Biological Sciences, Faculty of Science, University of Malaya, Kuala Lumpur, Malaysia
[2] College of Science and Technology, Wenzhou-Kean University, Wenzhou, Zhejiang, China
[3] Cancer Science Institute of Singapore, National University of Singapore, Singapore, Singapore
[4] National Wildlife Forensic Laboratory, Department of Wildlife and National Parks (PERHILITAN), Kuala Lumpur, Malaysia
[5] Centre for Research in Biotechnology for Agriculture (CEBAR), University of Malaya, Kuala Lumpur, Malaysia
[6] Department of Pathology, Faculty of Medicine, University of Malaya, Kuala Lumpur, Malaysia
[7] Zhejiang Bioinformatics International Science and Technology Cooperation Center, Wenzhou, Zhejiang, China
[8] Wenzhou Municipal Key Laboratory for Applied Biomedical and Biopharmaceutical Informatics, Wenzhou, Zhejiang, China
* These authors contributed equally to this work.

Corresponding author
Siew Woh Choo, cwoh@wku.edu.cn

## ABSTRACT

**Background:** The Malayan pangolin (*Manis javanica*) is a placental mammal and is listed as *Critically Endangered* on the IUCN Red List of Threatened Species. Most previous attempts to breed pangolins in captivity have met with little success because of dietary issues, infections, and other complications, although a previous study reported breeding pangolins in captivity to the third generation. In our previous pangolin genome sequencing data analysis, we obtained a considerable amount of bacterial DNA from a pregnant female Malayan pangolin (named "UM3"), which was likely infected by *Paraburkholderia fungorum*—an agent of biodegradation and bioremediation in agriculture.

**Methodology:** Here, we further confirmed and characterized this bacterial species using PCR, histological staining, whole-genome sequencing, and bioinformatics approaches. PCR assays with in-house designed primer sets and 16S universal primers showed clear positive bands in the cerebrum, cerebellum, lung, and blood of UM3 suggesting that UM3 might have developed septicaemia. Histological staining showed the presence of Gram-negative rod-shaped bacteria in the pangolin brain and lungs, indicating the colonization of the bacteria in these two organs. In addition, PCR screening of UM3's fetal tissues revealed the presence of *P. fungorum* in the gastrocnemius muscle, but not in other tissues that we examined. We also sequenced and reconstructed the genome of pangolin *P. fungorum*, which has a genome size of 7.7 Mbps.

**Conclusion:** Our study is the first to present detailed evidence of the presence of *P. fungorum* in a pangolin and her fetus (although preliminary results were presented

in our previous article). Here, we raise the concern that *P. fungorum* may potentially infect humans, especially YOPI (young, old, pregnant, and immunocompromised) people. Therefore, caution should be exercised when using this bacterial species as biodegradation or bioremediation agents in agriculture.

## INTRODUCTION

Pangolins are unique terrestrial mammals with special physical traits such as being covered in scales, lacking teeth, having poor vision, and having a well-developed sense of smell (*Ganguly, 2013*). Pangolins are extremely difficult to maintain and breed in captivity mainly because they frequently die of infections such as gastrointestinal infections, pneumonia, skin disease, parasitic infections, dietary issues, and other complications (*Yang et al., 2007*; *Hua et al., 2015*). During a previous pangolin genome sequencing project, we performed *in-silico* screening of the contig sequences for bacterial sequences, and found many exogenous DNA sequences (*Choo et al., 2016*).

The genus *Burkholderia* belongs to β-Proteobacteria—a Gram-negative, aerobic, rod-shaped bacteria associated with lethal human diseases (*Dias et al., 2019*). They are widely used in agriculture because they can fix nitrogen, promote plant growth, and degrade recalcitrant chemical compounds (*Dias et al., 2019*). They have been further categorized into *Burkholderia* and *Paraburkholderia*—the former being an animal and plant pathogen, and the latter being environmental and beneficial to plants (*Sawana, Adeolu & Gupta, 2014*; *Beukes et al., 2017*; *Estrada-de Los Santos et al., 2018*). Among the well-known members of *Burkholderia* are *B. pseudomallei* and *B. mallei* (both of which have been used as bioweapons in wars) (*Larsen & Johnson, 2009*; *Wheelis, 1998*), as well as *Paraburkholderia fungorum* which is a soil bacterium usually isolated from diverse ecological niches (*Coenye et al., 2001*; *Gerrits et al., 2005*). *Paraburkholderia fungorum* is commonly used in agriculture as a biodegradation and bioremediation agent (*Andreolli et al., 2011*; *Liu et al., 2019*). However, there have been reports of the isolation of *P. fungorum* from mouse nose and cystic fibrosis patients (*Coenye et al., 2001*). Therefore, there are debates on the suitability of *P. fungorum* in agricultural use because some people believe that it would affect human health, although no clear evidence has been reported (*Jones, Dodd & Webb, 2001*).

Here, we report a case of *P. fungorum* infection in a pregnant female pangolin (named "UM3") and its fetus, supported by evidence from polymerase chain reaction (PCR) assays, histological analysis, whole-genome analysis, and phylogenetic analysis. Our PCR and whole-genome sequencing results also showed the presence of this bacterial species in the muscles of the fetus, suggesting that *P. fungorum* may also have the capability to colonize the fetus.

## METHODS

### Ethics statement

Veterinary officers conducted all procedures involving dissecting animals under the oversight of experts from the Department of Wildlife and National Parks (DWNP) Peninsular Malaysia (PERHILITAN, Malaysia), following internationally recognized guidelines and approved by the University of Malaya Institutional Animal Care and Use Committee (reference number DRTU/11/10/2013/RH (R)).

### Biological samples

In 2012, more than 40 Malayan pangolins (*Manis javanica*) were seized in an anti-smuggling operation by PERHILITAN in Malaysia. One of the female pangolins ("UM3") was alive and pregnant, and weighed 2.73 kg at the time of confiscation. It was euthanized by the professional and experienced veterinarians at PERHILITAN using Dolethal® (Pentobarbitone sodium 200 mg/L; dosage: 1 mL/kg body weight) for animal welfare reasons. Only after the veterinary surgeon had pronounced the animal dead by checking for vital signs such as breathing and pulse, which ceased within 5 min of administering the drug, several organs including the cerebellum, cerebrum, lungs, thymus, liver, blood, heart, spleen and skin of the female and the fetus's cerebrum, cerebellum, intestine, kidney, cord blood, liver, lung, and gastrocnemius muscle were harvested and used in this study. The samples were immediately flash-frozen using liquid nitrogen and subsequently stored at −80 °C. For DNA extraction, minimal thawing was performed to obtain the tissue samples. Blood was drawn from seven additional live adults from the same seizure. Before conducting blood sampling, an anaesthetic mixture of ketamine-xylazine 1:1 (dosage: 0.5–1 mL per pangolin) or Zoletil 100 (3–4 mg/kg) was injected intramuscularly (IM) into the adult pangolins (276T, 2T9, 12T, 2T2, UM1, UM2, and UM3) to minimize their intervention. Once the muscles had relaxed, blood sampling was carried out *via* the coccygeal vein located at the tail, using a 5 mL syringe and 21G needle.

### Genomic DNA extraction and library preparation

The fetal gastrocnemius muscle tissue was used for genomic DNA extraction using a Qiagen Genomic Tips 20/G kit (Qiagen, Hilden, Germany) according to the manufacturer's protocol.

Genomic DNA libraries were prepared with a fragment length of approximately 300 bp and sequenced using Illumina HiSeq 2000 following the manufacturer's sequencing protocol.

### Discovery of bacterial sequences in the pangolin genome

The genome of the female pangolin was sequenced using the Illumina HiSeq 2000 platform (*Choo et al., 2016*). The tissue-specific genome assemblies were generated by CLC Assembly Cell using sequencing data from pangolin brain (cerebrum and cerebellum), liver, and lung samples. The sequencing reads were searched against a bacterial nucleotide sequence database using BLASTN (*Mount, 2007*). We screened the bacterial identity using

two criteria: 90% sequence identity and 90% sequence coverage, and 97% sequence identity and 97% sequence coverage.

## Average nucleotide identity (ANI) and average amino acid identity (AAI) analyses

The average nucleotide identity (ANI) values between bacterial species were calculated using previously described methods (*Goris et al., 2007*; *Rodriguez-R & Konstantinidis, 2014*; *Ang et al., 2016*; *Tan et al., 2016*). We used two-way BLAST and only used the forward and reverse-matched orthologs in the calculations. For robustness, the BLAST hits were filtered for at least 50% identity at the nucleotide and amino acid level, and a sequence coverage of at least 70%.

The protein sequences of 18 genomes belonging to the *Paraburkholderia* and *Burkholderia* genera were annotated using Rapid Annotation Search Tool (RAST) (*Aziz et al., 2008*). The RAST-predicted protein sequences for each assembly were retrieved, and average amino acid identity (AAI) values were calculated using the AAI calculator (*Goris et al., 2007*; *Rodriguez-R & Konstantinidis, 2014*).

## PCR assays

To further validate the presence of bacterial sequences in the pangolin, the frozen tissue samples of pangolin UM3 and her fetus were examined. Genomic DNA was extracted from nine adult tissues (cerebrum, cerebellum, liver, lungs, heart, spleen, thymus, skin, and blood) and eight fetal tissues (cerebrum, cerebellum, intestine, kidney, cord blood, liver, lung, and gastrocnemius muscle), and these were screened using polymerase chain reaction (PCR) assays. Three different target genomic regions that showed top hits to the bacteria identified from the previous BLASTN results were selected to design and synthesize novel PCR primers. We used bacterial universal 16S primers and three in-house designed primer sets (Table S1), targeting bacterial 16S rRNA, *Burkholderia*-specific transposase genomic region, OI25_7129 hypothetical protein genomic region, and the *P. fungorum*-specific DNA polymerase genomic region, respectively.

All PCR assays were performed using a total reaction volume of 50 μL containing 160 ng purified organ gDNA, 0.3 mol of each primer, deoxynucleotide triphosphates (dNTP, 400 μM each), 1.0 U Taq DNA polymerase and a supplied buffer. The PCR was performed as follows: one cycle (94 °C for 2 min) for initial denaturation; 35 cycles (98 °C for 10 s; 68 °C for 3 min) for annealing and DNA amplification. After completion of PCR, we visualized the PCR products on a 1% TAE agarose gel at 100 V for 70 min. The PCR products were purified by GeneJET PCR Purification Kit and directly sequenced with the same primers using BigDye© Terminator v3.1 Cycle Sequencing Kit (Applied Biosystems, Waltham, MA, USA) for validation.

## Tissue preparation and histological staining

We examined the histology of the adult pangolin's cerebellum and lungs. Each of the thawed organs was excised into two sets of smaller tissue pieces and fixed in 10% buffered formalin at 12 °C for a week, followed by embedding in paraffin wax to produce paraffin
blocks. For histology, the tissue blocks were sectioned on a rotary microtome (Leica RM2235; Leica Biosystems, Wetzlar, Germany) using a 3 μm blade. To prevent cross contamination, the blades were cleaned with 99% ethanol between sections. Subsequently, the slices were dewaxed using graded alcohol. Tissue slices were separately counterstained using hematoxilin/eosin (HE; Sigma, St. Louis, MO, USA) for tissue abnormality such as inflammation, and Brown-Hopps Gram stains for bacterial presence, as described by *Brown & Hopps (1973)*. Slices were examined under a light microscope with a Leica DF300 camera.

### Assembly of *P. fungorum* genomes

To further analyse the genomes of *P. fungorum*, we assembled genomes using three different strategies: (i) mapping reads from UM3's cerebrum and cerebellum whole-genome data onto *P. fungorum* reference genome ATCC BAA-463 (accession number: CP010024–CP010027) and assembling them into contigs; (ii) sequencing the DNA extracted from UM3's fetal muscle and mapping these reads onto the *P. fungorum* reference genome and assembling them into contigs, and (iii) mapping reads from UM3's cerebrum, cerebellum, and fetal muscle whole-genome data onto the *P. fungorum* reference genome and assembling them into contigs.

The core-genome single nucleotide polymorphisms (SNPs) were employed in constructing a sturdy phylogenetic tree to determine the taxonomic classification of these genomes. Core-genome SNPs are found in the core genome of a species or a group of closely related strains (*Vignal et al., 2002*) and are commonly used in phylogenetic studies to infer evolutionary relationships among bacterial populations. The use of core-genome SNPs has been shown to be a powerful tool for phylogenetic analyses, as they are less likely to be subject to homoplasy (convergence or reversal of nucleotide changes) compared to other markers. Seventeen *Burkholderia* and *Paraburkholderia* whole-genome sequences were retrieved from NCBI (http://www.ncbi.nlm.nih.gov) (*Pruitt, Tatusova & Maglott, 2007*; Table S2), with the sequences used being similar to those used by *Tan et al. (2020)*. The newly generated and reference sequences were uploaded to the PanSeq server to identify the core-genome SNPs in common genomic regions (*Laing et al., 2010*), and the extracted core SNPs were subsequently merged into a continuous sequence for each genome. Recent studies used conserved sequence indels (CSI) to study the relationships between species of *Burkholderia* and *Paraburkholderia* (*Sawana, Adeolu & Gupta, 2014*). Thus, CSI were used to further verify the taxonomic classification in this study. Conserved sequence indels were identified using the protein sequence of the assembled genome (iii) and 17 closely related species that were detected by ProteinOrtho (*Lechner et al., 2011*). The resultant 27 CSI were aligned using ClustalW (*Thompson, Gibson & Higgins, 2003*). The phylogenic trees using sequences of core genome SNPs and CSI were reconstructed using MEGA-X (*Kumar et al., 2018*). Neighbour-joining trees were inferred using the Kimura's two parameter model and nodal support was estimated using 1,000 non-parametric bootstrap replicates.

## RESULTS

### Presence of bacterial sequences in UM3

In our previous pangolin genome sequencing project, we sequenced the genomes of pangolin cerebellum, cerebrum and liver using the Illumina HiSeq2000 platform (*Choo et al., 2016*). During an *in-silico* bacterial sequence screening of the contig sequences of the tissue-specific assemblies using BLAST, we found many exogenous DNA sequences. The bacterial sequences were found in the assemblies of the cerebrum and cerebellum, but not in the liver assembly (Table S3). Specifically, in the assembled cerebral genome, there were 6,730 contigs mapped to bacterial genomes, where 6,635 of them (98.58%) had best matches with *P. fungorum*. Similarly, in the cerebellum-specific genome, 3,533 contigs mapped to bacterial genomes, among which 3,452 (97.7%) were from *P. fungorum*. These results indicate that the cerebrum and cerebellum tissues were predominantly colonised or infected by *P. fungorum* even though they should be sterile.

### PCR screening and sanger sequencing across different tissues of UM3

Our results showed clear positive bands in the lung, cerebrum, cerebellum, and blood of UM3 (Fig. 1). A weak positive band was also visible in the liver whereas no clear bands were observed in the other tissue types (Fig. 1).

### Histological examinations

To further confirm the presence of *P. fungorum* in UM3, the lung and cerebellum tissues were dissected and stained using Brown-Hopps Gram stains. Our staining revealed the presence of gram-negative and rod-shaped bacteria with a size of approximately 6–7 microns, supporting the notion that the lungs and cerebellum were invaded by *P. fungorum* (Figs. 2A, 2B, 2D, and 2E).

Histopathology screening was also performed using hematoxylin/eosin (H&E) staining to confirm *P. fungorum* infection in the lung and brain (cerebrum and cerebellum). The histological presentation of *Paraburkholderia* infection observed from lung tissues in other mammals is an abscess composed of cellular debris, numerous degenerate neutrophils, and macrophages that contain abundant intracytoplasmic basophilic material composed of rod-shaped bacteria (*Glaros et al., 2015*). However, our investigations showed no significant pathological signs in the dissected organs (Figs. 2C and 2F).

### Presence of *P. fungorum* in other adult pangolins

To examine whether the presence of *P. fungorum* in the UM3 pangolin organs was an isolated case, we randomly selected and screened the blood of four live adult pangolins (26T, 2T9, 12T, and 2T2) that were seized in the same batch as UM3. We also tested the blood of two live adult pangolins (UM1 and UM2) seized in a separate operation. UM3 was included as a positive control. All samples were screened for the presence of *P. fungorum* using the same primer sets as were used for UM3 and her fetus (Table S1). Of the seven blood samples, four (2T9, 12T, 2T2, and UM3) showed positive PCR bands for *P. fungorum*, and the bacterial identity was confirmed by Sanger sequencing and

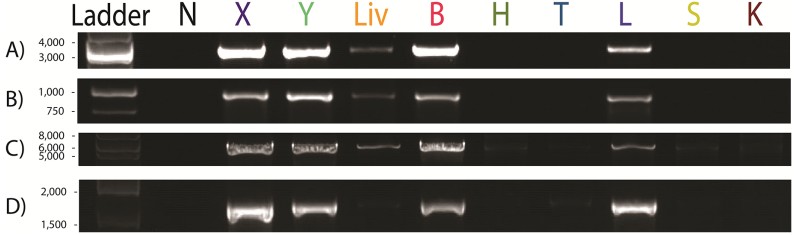

**Figure 1** *Paraburkholderia fungorum* **screening of different tissue types of the pregnant female Malayan pangolin (*Manis javanica*) UM3 using PCR assays.** Nine pangolin tissues and four primer sets were used, including PCR results for (A) *Burkholderia*-specific transposase genomic region, (B) OI25_7129 hypothetical protein genomic region, (C) *P. fungorum*-specific DNA polymerase genomic region, and (D) the bacterial universal 16S primer set. First lane is the negative control. N, Negative control; X, cerebrum; Y, cerebellum; Liv, liver; B, blood; H, heart; T, thymus; L, lung; S, spleen; K, kidney.

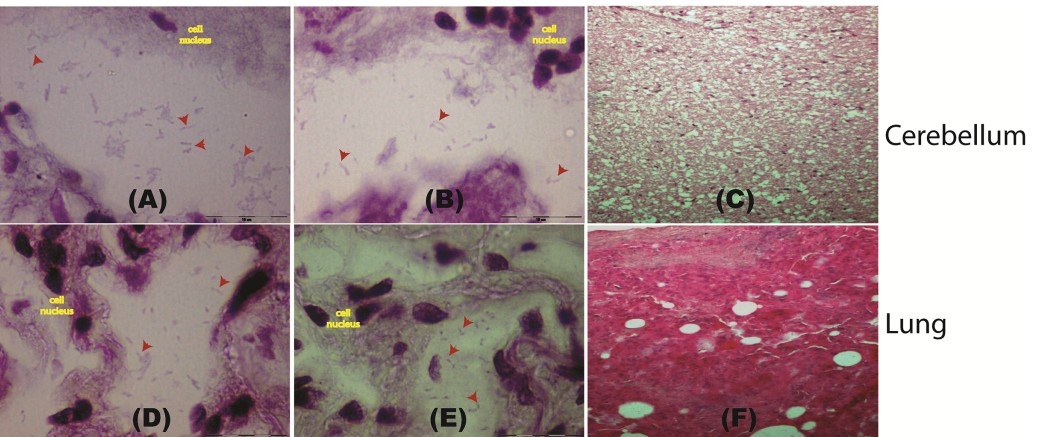

**Figure 2** **Histological staining of tissue samples obtained from a pregnant female Malayan pangolin (*Manis javanica*).** Staining results at a magnification of 100× for brain tissue (A–C) and lung tissue (D–F). Red arrows point to the locations of gram-negative rod-shaped bacteria. (C and F) Hematoxylin and eosin histological staining.

phylogenetic analyses (see *Choo et al., 2020*; Fig. 4 and Table S4), again indicating that the pangolins' blood was infected by *P. fungorum* (Fig. 3).

### *Paraburkholderia fungorum* may have the capability to infect the fetus

Since UM3 was pregnant, we wondered whether its fetus was also infected by *P. fungorum*. To examine this, we harvested and screened tissues from its fetus (including cord blood, lungs, intestine, kidney, liver, and brain) by performing PCR with the primer sets. We found that the fetal gastrocnemius muscle showed clear positive PCR bands (Fig. 4), and the bacterial identity was confirmed by Sanger sequencing and phylogenetic analyses (see *Choo et al., 2020*; Fig. 4 and Table S4). No significant bands were observed in other tissues (cerebrum, cerebellum, kidney, lung, cord blood, intestine and liver; Fig. 4). Furthermore, the fetal muscle genome was also sequenced using the Illumina HiSeq 2000 platform with a 20X sequencing coverage (after removing the pangolin sequences) and we found a substantial amount of *P. fungorum* DNA sequences.

**Figure 3 PCR assays of the blood of seven adult Malayan pangolins (*Manis javanica*).** UM1 and UM2 were seized in one operation, whereas UM3, 26T, 2T9, 12T and 2T2 were seized together in a separate operation. N, negative control; UM3, positive control; Target A, *Burkholderia*-specific transposase genomic region; Target B, OI25_7129 hypothetical protein genomic region; Target C, *P. fungorum*-specific DNA polymerase genomic region; 16S, universal 16S bacterial primers.

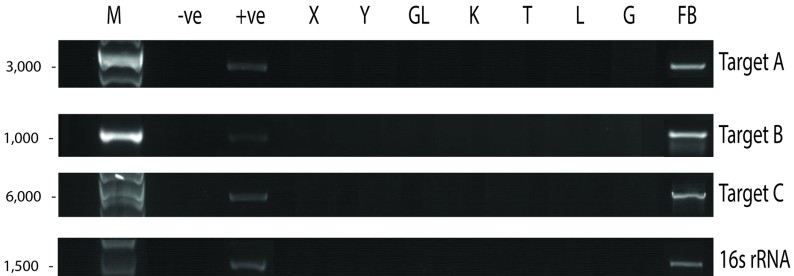

**Figure 4 *Paraburkholderia fungorum* screening of fetal Malayan pangolin (*Manis javanica*) tissue using PCR assays.** Target A, *Burkholderia*-specific transposase genomic region; Target B, OI25_7129 hypothetical protein genomic region; Target C, *P. fungorum*-specific DNA polymerase genomic region; 16S rRNA, universal 16S bacterial primers (−ve, negative control; +ve, positive control; X, cerebrum; Y, cerebellum; GL, intestine; K, kidney; T, cord blood; L, liver; G, lungs; FB, gastrocnemius muscle).

## Assembly of *P. fungorum* genomes

The newly assembled genome sequence of *P. fungorum* (accession numbers: CP028829–CP028832) can be downloaded from the GenBank database. Our

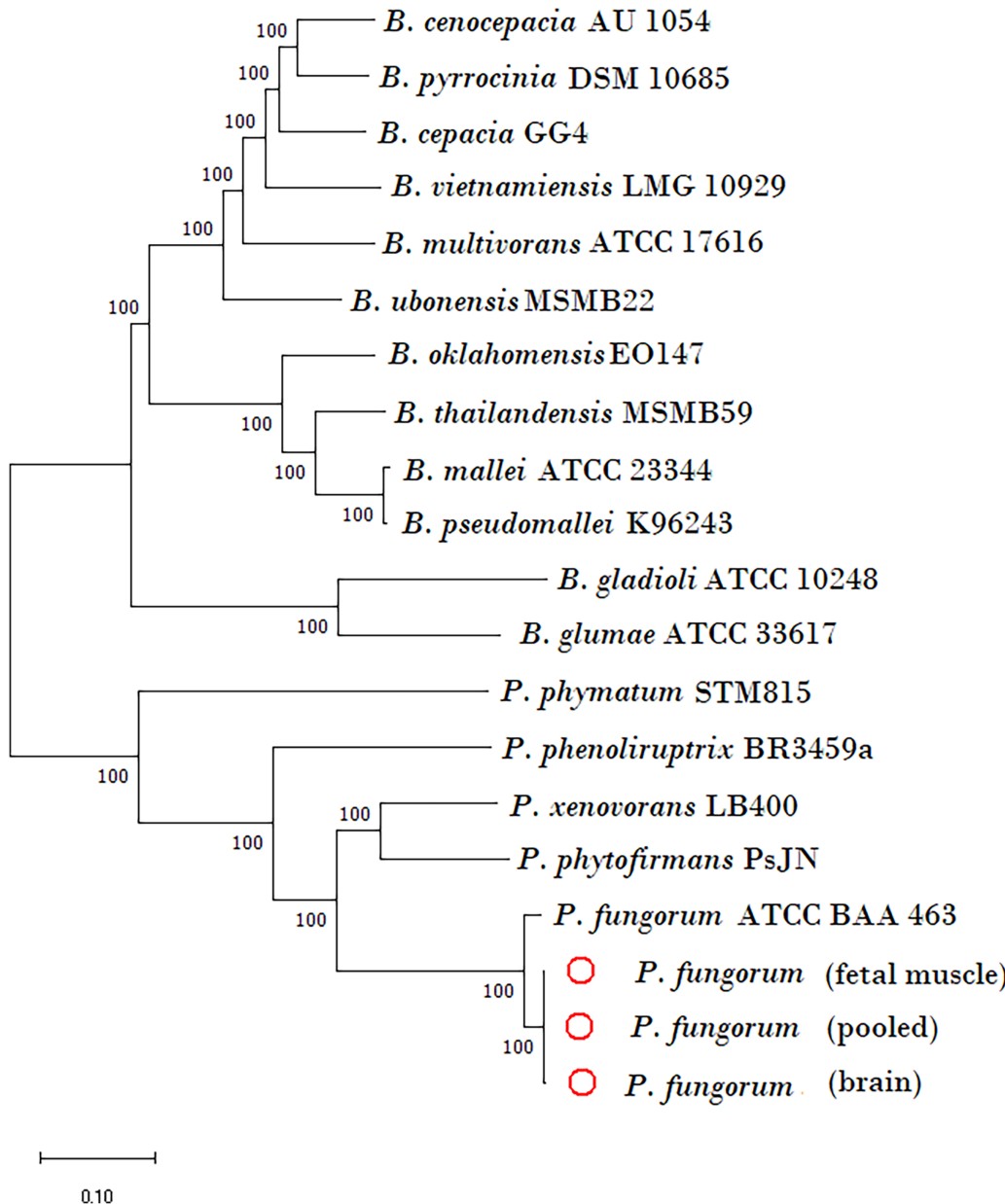

**Figure 5 Phylogenetic tree generated using core-genome SNPs mined from whole genomes of *Paraburkholderia fungorum* isolated from a Malayan pangolin (*Manis javanica*) fetus.** *Paraburkholderia fungorum* assemblies generated from fetal muscle, brain (cerebrum and cerebellum) and fetus pooled sequencing data (cerebrum, cerebellum and fetal muscle) aligned with the core-genome SNPs mined from genome sequences of 17 Burkholderial species. The phylogenetic tree was generated using the neighbour-joining (NJ) algorithm and 1,000 non-parametric bootstrap replications.

whole-genome results confirm that the sequences are most similar to the *P. fungorum* reference sequence ATCC BAA-463 (Figs. 5 and 6) which further validates the assignment of our sequences to *P. fungorum*.

To further validate this species assignment, we compared the *P. fungorum* genomes to other *Burkholderia* and *Paraburkholderia* species using ANI and AAI values. These

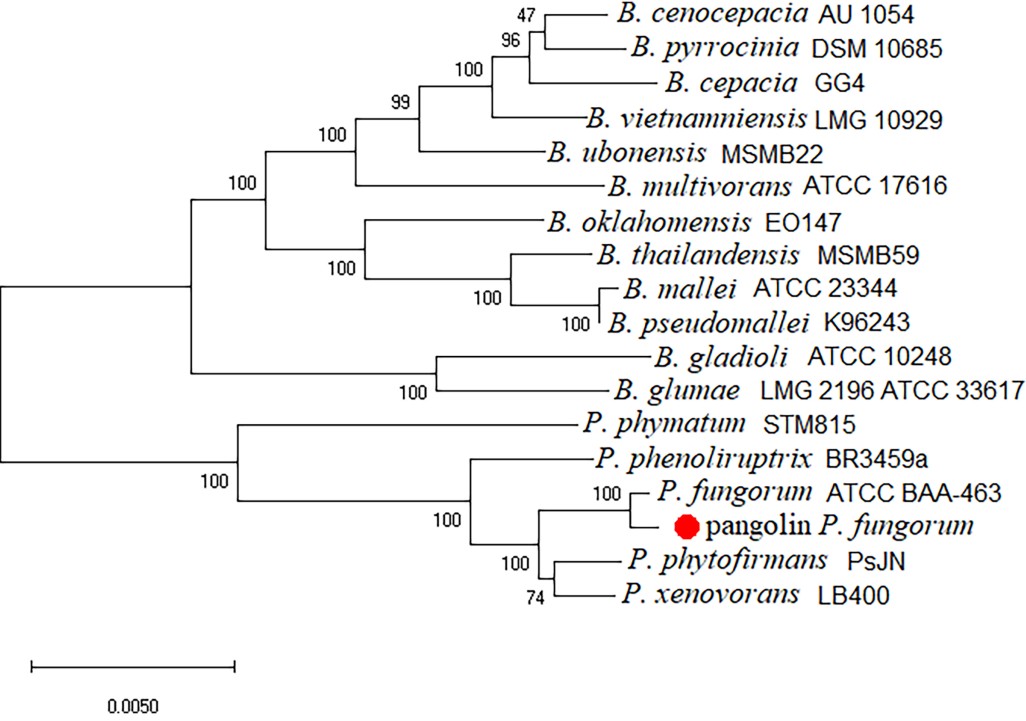

**Figure 6 Burkholderial phylogenetic tree generated using conserved proteins isolated from a Malayan pangolin (*Manis javanica*).** The conserved protein-based phylogenetic tree was generated using the neighbour-joining (NJ) algorithm and 1,000 non-parametric bootstraps replications.

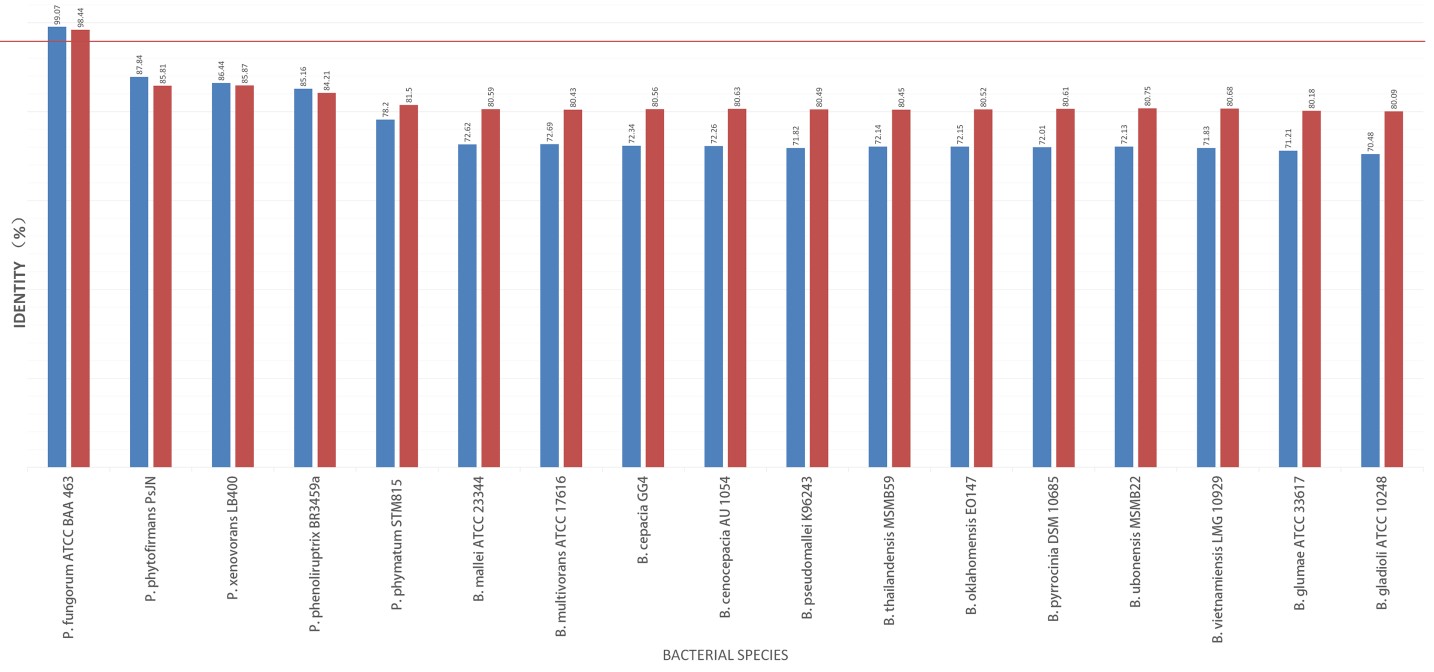

**Figure 7 Average nucleotide identity (ANI) and average amino acid identity (AAI) analyses of Burkholderial sequences isolated from Malayan pangolin (*Manis javanica*).** The horizontal red line indicates the 97% threshold above which sequences are deemed to belong to the same species. Blue bars represent AAI values, while marron bars represent ANI values.

comparisons indicated that the identified *P. fungorum* was closely related to the reference *P. fungorum* ATCC BAA-463, with an ANI value of 98.49% (Fig. 7). Other species had ANI values below the threshold of 97% used to define a species (*Goris et al., 2007*). The identified *P. fungorum* found in UM3's brain and fetal muscle had almost identical AAI and ANI values, indicating that they were from the same source. The ANI, AAI, and core-genome SNP-based phylogenetic analyses provided consistent evidence for the presence of *P. fungorum* in UM3.

## DISCUSSION

Here, we report a case of infection of *P. fungorum* in a pregnant placental mammal, the Malayan pangolin and its fetus. The presence of *P. fungorum* in Malayan pangolins was confirmed by PCR assays, histological examinations, whole-genome sequencing, phylogenetic analysis, ANI and AAI analyses. We detected *P. fungorum* in the cerebrum, cerebellum, lung, blood, and liver of the pregnant adult Malayan pangolin, but not in other tissues that we examined. Gram-negative and rod-shaped bacteria with a size of approximately 6–7 microns in the lungs and cerebellum provide strong evidence to support the invasion of *P. fungorum* in these supposedly sterile mammalian tissues. *Wiersinga et al. (2006)* suggested that the lung is the primary target organ for infectious *Burkholderia* species such as *B. pseudomallei* and *B. mallei*. Moreover, *P. fungorum* has been isolated from blood of humans with septicaemia, and the bacterium was being transported in the circulatory system to other host organs (*Coenye et al., 2001*; *Gerrits et al., 2005*). The fact that the adult female pangolin's blood tested positive for *P. fungorum* could indicates that she had developed septicaemia. Therefore, *P. fungorum* might have initiated a systemic infection through the lungs and spread to other critical organs, including the brain and blood.

Our results are the first to indicate that *P. fungorum* can colonize brain tissues. Colonisation of the brain could have occurred by the bacteria invading the blood-brain barrier (BBB), or through gaining access *via* the olfactory nerve. The second possibility is more likely for two main reasons. First, it has been demonstrated that other *Burkholderia* species such as *B. pseudomallei* can invade the nerves of the nasal cavity by colonizing the thin respiratory epithelium and rapidly migrating along the underlying trigeminal nerve to penetrate the cranial cavity, thus leading to direct brain infection without going through the BBB (*John et al., 2014*). Therefore, it is possible that the genetically related *P. fungorum* may also invade the pangolin brain *via* olfactory nerve cells. Second, pangolins are presumed to have weak immunity due to the loss of the interferon epsilon (IFNE) gene, which is exclusively expressed in other mammalian epithelial cells and is important for skin and mucosal immunity (*Choo et al., 2016*). The weakened mucosal immunity of pangolin may make the invasion of *P. fungorum* into the olfactory epithelium easier.

Altogether, our data have confirmed the existence of *P. fungorum* in the fetal muscle and suggest the possibility of transplacental infection or an ascending infection pathway from the cervix. Our PCR assays showed the presence of *P. fungorum* in pangolin fetal gastrocnemius muscle, but not in other fetal tissues (*e.g.*, cord blood, lung, and brain). Our results suggest that this bacterial species can infect a mammalian fetus *in vitro*;

however, its underlying mechanism remains unknown. Notably, *B. pseudomallei* has been reported to cause infectious disease in a pregnant woman, resulting in intrauterine infection with a subsequent spontaneous abortion (*Chang et al., 2020*). Therefore, it is possible that pangolin *P. fungorum* can colonized the fetal muscle *via* a transplacental invasion as previously shown in *B. pseudomallei* in goats (*Choy et al., 2000*). However, this scenario is unlikely since we could not detect *P. fungorum* in the umbilical cord blood. Another possibility is that *P. fungorum* invaded the fetus through the urinary tract. This mechanism is similar to the invasive Group B *Streptococcus* bacteria that are able to infect the perinatal space in humans (*Cagno, Pettit & Weiss, 2012*). Another possible mechanism is that *P. fungorum* may be an invasive bacterial species that can penetrate the mucosa-protected cervix of the female and bypass the amnion of the uterus and the fetal skin before arriving in the leg muscle. Notably, some invasive pathogens such as Group B *Streptococcus* (*Cagno, Pettit & Weiss, 2012*), *Listeria monocytogenes* (*Gibbs & Duff, 1991*) and *Mycoplasma hominis* (*Eschenbach, 1993*) are known to use this route to infect fetuses. If it is true, this could be the first indirect evidence to show that *P. fungorum* can be an invasive bacterial species, and this possibility deserve further study.

*Paraburkholderia fungorum* was detected in more than half of the seven individuals that were tested, indicating that it is not an isolated case. All the pangolins that tested positive originated from the same seizure. It is possible that *P. fungorum* was transmitted between the individuals in this seizure due to their unnaturally close proximity and probably compromised immunity owing to the stress of being trafficked. This possibility is lent support by the phylogenetic analyses and sanger sequencing, which confirmed that they carried identical *P. fungorum* sequences. Based on these limited observations, animal-to-animal transmission may be common in seized pangolins, perhaps due to their reported poor immunity (*Hua et al., 2015*; *Choo et al., 2016*) especially under stressful conditions. The lack of obvious pathological manifestations in the tissues examined in this study despite the confirmed presence of *P. fungorum* also merits further study. The lack of pathological symptoms may suggest that the bacteria is able to colonise pangolin tissues without any overt symptomatic presentation, although we cannot rule out the possibility that the infection was at an early stage, with a resultant absence of any observable histological changes in the tissues. Therefore, we believe that our findings suggest the need for increased vigilance and testing for diseases in captive pangolins, particularly those that have been subjected to stressful conditions.

In light of the Malayan pangolin's ecology of burrowing in the soil, sleeping in burrows and foraging in ant and termite nests, they could probably obtain *P. fungorum* naturally from the environment (*Gray et al., 2023*; *Lim & Ng, 2008*). However, all tested pangolins were from the illegal wildlife trade, where they would be stressed, maintained in an unnatural environment, and occur in unnaturally close proximity for an unnaturally long period of time. It is generally accepted that trafficking (and the resultant reduction in immunity brought on by stress) increases the risk of disease spillover between species (*Tajudeen et al., 2022*). Therefore, the fact that *P. fungorum* was found in pangolins in trade does not necessarily mean that it naturally occurs in pangolins. Thus, the

investigation of wild pangolins needs to be undertaken to assess whether this is an example of a spillover event, or whether *P. fungorum* does indeed occur naturally in wild pangolins.

In another aspect, several other opportunistic pathogenic burkholderial species (*e.g.*, *B. phytofirmans* and *B. cepacia* complex; *Andreolli et al., 2011*; *Estrada-de los Santos et al., 2013*) have been suggested as bioremediation/ biodegradation agents for polycyclic aromatic hydrocarbon (PAHs) contaminated soil (*Andreolli et al., 2011*) and oxidised halo-benzene contaminated water (*Dobslaw & Engesser, 2015*; *Strunk & Engesser, 2013*). *Paraburkholderia fungorum* is able to degrade the PAH phenanthrene, as well having the ability to remove heavy metals from contaminated soil (*Liu et al., 2019*).The use of *Burkholderia* species including *P. fungorum* in bioremediation, however, potentially increases the possibility of burkholderial infection in both humans and animals by artificially introducing these bacteria into the environment and should be treated with caution. Similarly, the extensive use of Paraburkholderiales as a plant growth promoting bacteria (PGPB) and plant growth promoting rhizobacteria (PGPR) in agriculture needs to be revised and re-evaluated. However, our previous study has demonstrated the presence of virulence and defence mechanisms associated with pathogenesis in the pangolin genome data (*Tan et al., 2020*) as well as a histopathological distribution in organs supporting its pathogenicity in pangolins. Taken together with other documented cases of *P. fungorum* in humans and animals (*Gerrits et al., 2005*; *Loong et al., 2019*; *Nally et al., 2018*), we posit that this species could be classified as a potential and probably opportunistic pathogen. *Burkholderia* species exhibit zoonotic capabilities as well as being opportunistic pathogens (*Elschner et al., 2014*), however the zoonotic capabilities of the subgroup *Paraburkholderia* is not yet well understood. Hence, the results of this study identifying *P. fungorum* in pangolins supports the possibility of its zoonotic and opportunistic potential. This is especially so as the human pathogenic species has also previously been isolated in the cerebrospinal fluid (*Coenye et al., 2001*) and synovial tissue of humans (*Loong et al., 2019*). Notably, there are some reported clinical cases such as a 9-year-old female with *P. fungorum* causing septicaemia (*Gerrits et al., 2005*), a 66-year-old woman with *P. fungorum* observed in the cerebrospinal fluid (*Wiersinga et al., 2006*), and *P. fungorum* cultured from a pregnant woman's vaginal secretion (*Wiersinga et al., 2006*). Our study showed that *P. fungorum* could cause septicaemia and colonize the brain and lungs, as well as fetus, supporting the pathogenicity of *P. fungorum*. Our study may raise an alert on the use of *P. fungorum* in agriculture. We cannot rule out the possibility that *P. fungorum* may potentially target YOPI (young, old, pregnant, and immunocompromised) people.

Our study highlights the importance of improving the management of these endangered pangolins in captivity. Careful treatment and extensive medical care should be provided to pangolins in captivity because they frequently succumb to infection. It is important to provide a hygienic environment (as well as hygienic food and water) when keeping pangolins in captivity in order to minimize the risk of infection and stress. Regular monitoring of possible infections (*e.g.*, blood tests if individuals show signs of disease) may also be an important measure in the rescue and conservation of pangolins in captivity.

## CONCLUSION

This study provides insight into the first discovery of *Paraburkholderia fungorum* in the Malayan pangolin. We believe that pangolins can be a reference for humans, particularly immunocompromised people, due to their reduced immunity. Our study may also raise concern over the usage of *P. fungorum* as biodegradation or bioremediation agents in agriculture. Limited information is available in the literature regarding the potential impacts of this bacteria on pangolin health and conservation. However, given the importance of pangolin conservation and the threat of disease to their survival, further research is needed to understand the potential risks posed by *P. fungorum* to this *Critically Endangered* species. More research is necessary to determine the potential transmission pathways of *P. fungorum*, the effects of exposure to the bacteria on pangolin health, and potential management strategies to mitigate the risk of transmission.

### Funding

This work was funded by the High-Level Talent Recruitment Programme for Academic and Research Platform Construction (Reference Number: 5000105) from Wenzhou-Kean University. Furthermore, Ka Yun Tan was supported by the Centre for Research in Biotechnology for Agriculture (CEBAR) grant IRU-MRUN (RU023-2015). The funders had no role in study design, data collection and analysis, decision to publish, or preparation of the manuscript.

### Grant Disclosures

The following grant information was disclosed by the authors:
Wenzhou-Kean University: The High-level Talent Recruitment Programme for Academic and Research Platform Construction: 5000105.
Centre for Research in Biotechnology for Agriculture (CEBAR): IRU-MRUN (RU023-2015).

### Competing Interests

The authors declare that they have no competing interests.

### Author Contributions

- Ka Yun Tan performed the experiments, analyzed the data, prepared figures and/or tables, authored or reviewed drafts of the article, and approved the final draft.
- Siwei Deng performed the experiments, analyzed the data, prepared figures and/or tables, authored or reviewed drafts of the article, and approved the final draft.
- Tze King Tan performed the experiments, analyzed the data, authored or reviewed drafts of the article, and approved the final draft.
- Ranjeev Hari performed the experiments, analyzed the data, authored or reviewed drafts of the article, and approved the final draft.

- Frankie Thomas Sitam performed the experiments, authored or reviewed drafts of the article, and approved the final draft.
- Rofina Yasmin Othman performed the experiments, analyzed the data, authored or reviewed drafts of the article, and approved the final draft.
- Kum Thong Wong performed the experiments, analyzed the data, authored or reviewed drafts of the article, and approved the final draft.
- Taznim Begam Mohd Mohidin authored or reviewed drafts of the article, and approved the final draft.
- Siew Woh Choo conceived and designed the experiments, authored or reviewed drafts of the article, and approved the final draft.

## Animal Ethics

The following information was supplied relating to ethical approvals (*i.e.*, approving body and any reference numbers):

Veterinary officers conducted all procedures involving animals and experts at the Department of Wildlife and National Parks (PERHILITAN), Malaysia, following internationally recognized guidelines and approved by the University of Malaya Institutional Animal Care and Use Committee (UM IACUC) [reference number of the approval: DRTU/11/10/2013/RH (R)].

## Data Availability

The genomic data is availalble at GenBank: CP028829–CP028832.

The *P. fungorum* ATCC BAA-463 reference genome is also available at GenBank: CP010024–CP010027.

## Supplemental Information

Supplemental information for this article can be found online at http://dx.doi.org/10.7717/peerj.16002#supplemental-information.

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
