# Peer review of "Genome sequence analysis of Malayan pangolin (Manis javanica) forensic samples reveals the presence of Paraburkholderia fungorum sequences"

_PeerJ, doi:10.7717/peerj.16002_

## Round 0.1 · original submission · Major Revisions

Most reviewers are suggesting major revisions. Hence my recommendation to revise the manuscript.

·

Basic reporting

This is an interesting study that provides additional details on a result that was partially mentioned in a previous publication (Tan, K.Y., Dutta, A., Tan, T.K., Hari, R., Othman, R.Y. and Choo, S.W., 2020. Comprehensive genome analysis of a pangolin-associated Paraburkholderia fungorum provides new insights into its secretion systems and virulence. PeerJ, 8, p.e9733.). The contents are worthy of publication, however I believe that this manuscript requires quite a bit of revision before it can be accepted for publication.

In general the language was acceptable, although I have made some suggestions where this can be improved.

The structure of the article needs to be worked on. There are several instances where text that describes the methods used, and which therefore should be included in the Materials and Methods section, are included in the Results. Similarly, there are instances in the Results section where the results are interpreted and discussed - these interpretations need to be moved to the Discussion. Only present your results in the Results section.

The figure captions also require some attention. Figure captions need to be "stand-alone" captions, i.e., if someone only sees the figure and the caption (without seeing the remainder of the article), they still need to be able to understand the figure. To do so, you need to include more information in many of your figure legends.

I have made some suggestions in the annotated manuscript in these regards.

Experimental design

In general, the experimental design appears to be appropriate and the authors explicitly state how their research fills an identified knowledge gap.

There are some shortcomings in the reporting of the experimental design, however. In many instances, insufficient detail is provided to enable anyone to repeat the analyses. This is particularly evident in the phylogenetic analyses, where none or very little mention is made of what models were used to infer the phylogenies, whether bootstrapping was done (and if so, how many replicates), and what model of species evolution was used and how this model was selected. Insufficient detail is also provided on what tissue types were sampled from the adult female pangolin, and what tissue types were sampled from her fetus. In addition, more details need to be provided regarding the source of these pangolins, particularly whether they were alive or dead when received, and if dead how they were stored and handled post mortem.

No information is provided on how the sequences for the other Burkholderial sequences that were used in the phylogenetic reconstructions were selected. Based on the results of Sawana et al. (2014), only a small proportion of the available sequences were used. How were the ingroup taxa selected? And where were these sequences sourced from? It would also be good to include the sequence accession numbers for all these reference sequences as supplementary material.

Validity of the findings

The findings in general appear appropriate and supported by the results, however as mentioned above some restructuring is necessary to make the findings and the conclusions clearer.

One aspect that I would like to see discussed, even if briefly, is the potential effect that the source of these pangolins could have on the results. The pangolins that were tested were all sourced from the illegal trade, where they would be stressed, maintained in an unnatural environment, and occur in unnaturally close proximity for an unnaturally long length of time. It is generally accepted that trafficking (and the resultant reduction in immunity brought on by stress) increases the risk of disease spillover between species. The fact that P. fungorum was found in pangolins in trade does not necessarily mean that it naturally occurs in pangolins. This is lent support by P. fungorum only being detected in individuals that were al sampled from the same trafficking event. A caveat needs to be added that investigations of wild pangolins needs to be undertaken to assess whether this is indeed an example of a spillover event, or whether P. fungorum does indeed occur naturally in wild pangolins.

The authors' conclusion that their results suggest that increased vigilance of, and testing for, diseases in captive pangolins is, in my opinion, largely unfounded. As mentioned above, just because P. fungorum was found in one consignment of trafficked pangolins it does not mean that all pangolins have P. fungorum. Furthermore, the interventions that the authors propose for captive pangolins raise both ethical and long-term health concerns for captive pangolins. I highly recommend that the authors do not extrapolate their results from a single trafficking event to all captive pangolins, as I do not believe that their results support such an inference.

Reviewer 2 ·

Basic reporting

Comment: I thoroughly reviewed the manuscript entitled “Genome sequence analysis of archival Malayan pangolin (Manis Javanica) tissues reveals the presence of Paraburkholderia fungorum sequences” by Tan et al. This manuscript provides information about the presence of P. fungorum in a pregnant mammalian pangolin species and a fetus. They raised the concern that P. fungorum potentially infect humans, especially YOPI (young, old, pregnant, and immunocompromised) people. Manuscripts is well written and highlighted the issue related to the importance of improving the conservation and breeding strategies of threatened and immunologically fragile mammal. Although the use of the methodology and analytical approaches seem appropriate, but there are some flaws in presentation of results author need to add suggestion.
1. In the line no 69, authors need to have add in details about how these bacteria affect the pangolin with proper citation. Because it is very crucial to pangolin conservation.
2. In the line no. 185-186, authors are not sure about pathological effect of bacteria on pangolin. Then authors need to justify the reason the present study.
3. Statement in the line no. 236 to 2 243 is not clear. Author needs to reframe.
4. In the line no 311-313 They also raised the concern about using P. fungorum as a biodegradation or bioremediation agent in agriculture. Please elaborate it.
5. Largaly, authors in This study Authors highlighted the discovery of Paraburkholderia genus fungorum species in a mammal species, Pangolin. They also mentioned that pangolin can be a reference for humans, particularly immunocompromised people, in studying P. fungorum infection. Here author need to highlight about how this study help in Pangolin Conservation in wild.

Experimental design

NA

Validity of the findings

NA

Additional comments

NA

Reviewer 3 ·

Basic reporting

Overall, an interesting read. However there are some parts that can be improved upon for better continuity and clarification, ie:

Title: Suggest a more relevant title to better reflect the content of the manuscript eg "Genome sequence analysis of Malayan pangolin (Manis javanica) forensic samples reveal the presence of Paraburkholderia fungorum sequences' as only 1 recently dead carcass was examined for the study.

Background:
Line 29: Suggest to edit for better clarification eg 'Previous efforts have attempted to breed pangolins in captivity, but with little success because of dietary issues, infections, and others'.
Line 30: Need to clarify that the tissues were obtained from a dead pregnant female Malayan pangolin.

Conclusion:
Line 43-44: Suggest to edit to 'the presence of P. fungorum in the carcass of a pregnant mammalian pangolin species and a fetus'.
Line 45-46: Suggest also to edit 'Therefore, caution should be exercised in using this bacterial species as....'

Introduction:
Line 55: Suggest editing to 'and strong sense of smell'
Line 56-57: need also add on other factors such as dietary issues etc
Line 64: need add references on Burkholderia spp. being used as bioweapons in wars?
Line 67: is it 'mouse nose' or 'mouse nostril'?

Experimental design

Ethics statement:
Line 79-80: need clarify what was meant as 'Veterinary officers conducted all procedures involving animals and experts at the Department of Wildlife and National Parks (DWNP), Malaysia'?
Line 79-80: Also corrections should be made to read as 'the Department of Wildlife and National Parks (DWNP) Peninsular Malaysia'.

Biological samples:
Line 85: Suggest to edit 'The samples of a dead pregnant pangolin...'
Need also some information on when these tissues were harvested, and when the study was conducted. Was studies on the samples conducted in Malaysia or out of Malaysia, and was permits obtained if the studies were conducted overseas in accordance with Malaysian laws and legislation?

Library preparation and sequencing
Line 96: to correct spacing between 'vendor's sequencing...'

Validity of the findings

Discussion:
The authors proved that Burkholderial spp was detected in the pangolin samples. However, there is no explanation on how the pangolins could have become infected? Looking at the Malayan pangolin's ecology and lifestyle of burrowing in the soil, sleeping in the burrows, tearing apart ant and termites' nests, would it be possible that the pangolins were infected naturally? Perhaps the authors can elaborate further on this possible route of infection.

Additional comments

Some grammatical errors spotted throughout the text. I would suggest the authors to recheck the manuscript for other possible errors.

---

## Round 0.2 · Minor Revisions

Suggestions made by one reviewer required revision. Hence my recommendation is to revise the manuscript and submit again.

·

Basic reporting

The language is, for the most part, suitable throughout. The manuscript is well laid out, and all relevant results, tables and figures are included. All appropriate literature appears to have been cited.

Experimental design

The experimental design is appropriate and has been satisfactorily executed. Sufficient information has now been included in the Methods section to make this research repeatable.

Validity of the findings

The conclusions drawn are supported by the presented results.

Additional comments

This manuscript is an improvement over the previous version, and I thank the authors for their efforts in addressing my previous comments. I have no major comments or concerns regarding this manuscript anymore.

I have made some suggested edits in the accompanying manuscript. These edits are mostly of a grammatical nature and are intended to improve the flow and clarity of the manuscript, without changing the meaning of any sentences. There are two instances where I moved sections from the Results to the Discussion, and one instance in the Discussion where I amalgamated two paragraphs that had very similar content.

Overall this is a very interesting, well-written article and I would like to congratulate and thank the authors for their efforts.

Reviewer 2 ·

Basic reporting

All raised points are addressed.

Experimental design

All experimental design are up to the mark.

Validity of the findings

valid

Additional comments

nil

---

## Round 0.3 · Minor Revisions

As reviewer suggested minor changes in the manuscript, hence my suggestion is to revise

·

Basic reporting

Overall, this is a very interesting and well-written manuscript, and I applaud the authors for implementing all of the reviewers' prior recommendations. I have made a few final suggested edits in the accompanying annotated manuscript, although most of these just refer to grammar conventions. There are two instances where it would be good to include a reference to support a statement, five instances where a bit more clarity is required, minor suggested edits to two of your figures, and a few other minor suggested edits.

Experimental design

The experimental design is appropriate and, with the exception of one minor omission which is highlighted in the accompanying manuscript, is reported on in sufficient detail.

Validity of the findings

The findings are valid, and all conclusions are supported by the current findings and/or previously published results which are appropriately referenced.

Additional comments

This is an interesting study and I look forward to seeing it published.

---

## Round 0.4 · accepted · Accept

The manuscript is scientifically sound and provides information about the Malayan pangolin. Hence, my recommendation is to accept.